# Discussion on the relationship between gut microbiota and glioma through Mendelian randomization test based on the brain gut axis

**Chenzhi Cui[1,2‡], Tianke Yang[3,4‡], ShengYu Wang[5‡], Zhuqiang Jia[6,7], Lin Zhao[8], Xin Han[7,9], Xiaohong Sun[10], Junwei Zong[3]\*, Shouyu Wang📧[3]\*, Dong Chen[1,2]\***

**1** Graduate school, Dalian Medical University, Dalian, Dalian, China, **2** Department of Neurosurgery, Dalian Municipal Central Hospital, Dalian, China, **3** Department of Orthopaedic Surgery, The First Affiliated Hospital of Dalian Medical University, Dalian, China, **4** College of Integrative Medicine, Dalian Medical University, Dalian, China, **5** Medical Laboratory Technology, College of Medical Laboratory, Dalian Medical University, Dalian, China, **6** The First Affiliated Hospital of Dalian Medical University, Dalian, China, **7** Naqu People's Hospital, Tibet, China, **8** Department of Quality Management, Dalian Municipal Central Hospital, Dalian, China, **9** Department of Orthopaedic Surgery, The Second Affiliated Hospital of Dalian Medical University, Dalian, China, **10** Department of Nursing, The First Affiliated Hospital of Dalian Medical University, Dalian, China

‡ CC, TY and SW contributed equally to this work and share first authorship.
* wangshouyu666@126.com (SW); aweizone@163.com (JZ); neuro-chen@dlut.edu.cn (DC)

**Data Availability Statement:** The data utilized in this research is publicly accessible and can be downloaded from the following online platforms or repositories:MiBioGen Consortium: https://

## Abstract

### Background

In the realm of Gut-Brain axis research, existing evidence points to a complex bidirectional regulatory mechanism between gut microbiota and the brain. However, the question of whether a causal relationship exists between gut microbiota and specific types of brain tumors, such as gliomas, remains unresolved. To address this gap, we employed publicly available Genome-Wide Association Study (GWAS) and MIOBEN databases, conducting an in-depth analysis using Two-Sample Mendelian Randomization (MR).

### Method

We carried out two sets of MR analyses. The preliminary analysis included fewer instrumental variables due to a high genome-wide statistical significance threshold ($5×10^{-8}$). To enable a more comprehensive and detailed analysis, we adjusted the significance threshold to $1×10^{-5}$. We performed linkage disequilibrium analysis ($R^2 <0.001$, clumping distance = 10,000kb) and detailed screening of palindromic SNPs, followed by MR analysis and validation through sensitivity analysis.

### Results

Our findings reveal a causal relationship between gut microbiota and gliomas. Further confirmation via Inverse Variance Weighting (IVW) identified eight specific microbial

molgenis26.gcc.rug.nl/downloads/MiBioGen/MBG.
allHits.p1e4.txt IEU Open GWAS project: https://
gwas.mrcieu.ac.uk/datasets/prot-a-1217/.

**Funding:** This study was supported by the National
Natural Science Foundation of China (82074426,
82104864, 82204822), Natural Science Foundation
of Liaoning Province (2021-BS-215, 2022-MS-25,
2023-MS-13), Liaoning Revitalization Talents
Program (XLYC1802014), Liaoning Key Research
and Development Planning Project (2017226015),
Basic Research Projects of Liaoning Provincial
Department of Education (LJKMZ20221286),
Naural Science Foundation of Tibet Autonomous
Region and Regional Science(XZ202301ZR0030G,
XZ2023ZR-ZY82(Z)) and Technology Project of
Naqu City. The funder provided writing ideas and
publication decisions for this article.

**Competing interests:** The authors have declared
that no competing interests exist.

**Abbreviations:** FDR, False Discovery Rate; GWAS,
Genome-wide association study; IVs, Instrumental
variables; IVW, Inverse variance weighting; LD,
Linkage disequilibrium; MR, Mendelian
randomization; MR-PRESSO, Mendelian
randomized polymorphism residual and outlier;
SM, Simple mode; SNPs, Single nucleotide
polymorphisms.

communities related to gliomas. Notably, the Peptostreptococcaceae and Olsenella communities appear to have a protective effect, reducing glioma risk.

## Conclusion

This study not only confirms the causal link between gut microbiota and gliomas but also suggests a new avenue for future glioma treatment.

## Introduction

Gliomas are the most lethal tumors in the Central Nervous System (CNS), accounting for a malignancy rate as high as 80% among brain tumors. Despite medical advancements in diagnosis and treatmentin central nervous system tumors, gliomas account for 23.3%, and in all malignant brain tumors, they make up 78.3%. Their five-year mortality rate ranks after pancreatic and lung cancers [1].They also have the highest mortality rate in this category [2].The incidence of gliomas is on the rise, becoming a significant global health concern. Adding to this, a disheartening mere 5.6% [3]. According to the World Health Organization's classification, gliomas vary in grades, each associated with different prognostic outcomes [4]. Grades 3 and 4 are considered high-grade gliomas, with a median survival time ranging from 15 months to 3 years [5].

Current cancer treatments are limited in their effectiveness against gliomas due to the blood-brain barrier [6, 7].While chemotherapy and radiation therapy can extend survival to some extent, they often come with severe side effects and a decline in quality of life. Moreover, the high likelihood of recurrence is due to the tumor's heterogeneity and invasiveness. Recent scientific research has unveiled complex bidirectional communication between the brain, gut, and their microbiomes [8].This opens up a new avenue for potentially influencing glioma development and treatment efficacy through gut microbiome regulation. The gut microbiome not only aids in digestion and nutrient absorption but also interacts closely with the immune, endocrine, and nervous systems [9–11]. Reports indicate valuable new insights into the role of the Gut-Brain axis in treating cancer and CNS diseases [12–15]. Our hypothesis is grounded in the impact of the gut microbiome on brain health and disease through various mechanisms and the mutual regulation of the " Gut-Brain axis " (Fig 1), supported by the latest discoveries in neuro-oncology and microbiology fields. For instance, changes in the gut microbiota are linked to alterations in immune factors and inflammation, both of which play crucial roles in tumor development.

In exploring the complex relationship between the gut microbiome and gliomas, Confounding factors encompass not only common variables such as age, gender, and underlying health conditions but also include lifestyle choices, socioeconomic status differences, psychological factors, genetic predisposition, and family environment. These factors cannot be overlooked in the impact they have on the outcomes of randomized controlled trials (RCTs) and clinical trials [16]. Observational studies and RCTs may not provide unbiased estimates of the true association between exposure and outcome when there are too many confounding factors [17]. Therefore, this study employs Mendelian randomization to further investigate any causal link between gliomas and the gut microbiome, providing robust scientific evidence for future clinical trials and treatment methods [18, 19]. Using Mendelian Randomization offers a significant advantage over other observational studies by leveraging genetic variations as instrumental variables to establish more reliable causal inferences, thus minimizing confounding factors. The gut microbiome exhibits resilience and resistance to change. Maintaining a beneficial

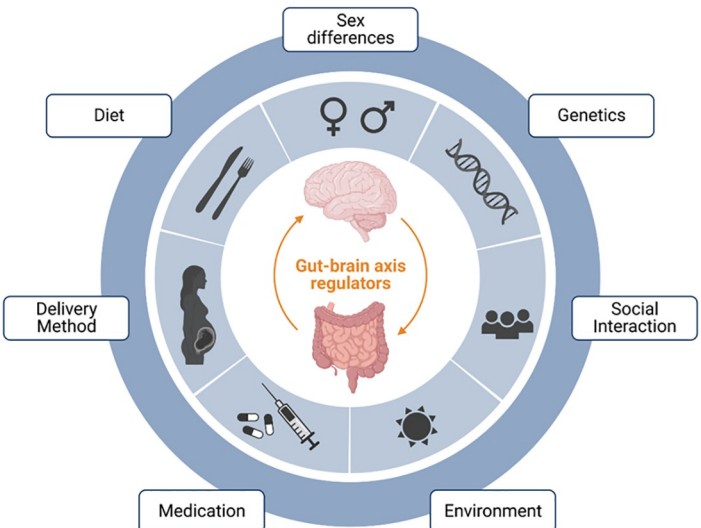

**Fig 1. The Gut-Brain axis is a complex bidirectional communication system that links the brain and the digestive system, affecting our emotions and health.** This interaction is influenced by a variety of internal and external factors, including but not limited to gender, genetics, social environment, medication use, dietary habits, and mode of childbirth.

microbial community requires a state of equilibrium both within the microbial population and at the interface between the microbes and the host's intestinal lining. The resilience of a healthy microbiome shields us from diseases related to ecological imbalance [20].

## Method

### Ethics approval and consent to participate

The data used in this study were all from public databases that can be downloaded directly for research purposes and do not involve the reporting or using of any animal, human, or tissue data.

### 1. Data sources

Then gut microbiome is an itegral part of human physiology [21]. The genetic variation data discussed in this article comes from MiBioGen(https://molgenis26.gcc.rug.nl/downloads/ MiBioGen/MBG.allHits.p1e4.txt), the largest global database for gut microbiome whole-genome meta-analysis [22].This database has collected 16S rRNA gene sequencing and geno-typing data from 18,340 participants across 24 countries, including Sweden, Finland, and the United Kingdom [22]. To analyze the composition of the microbiome, MiBioGen target the variable regions V4, V3-V4, and V1-V2 of the 16S rRNA gene, applying direct taxonomic binning for classification. We conduct microbial quantitative trait locus (mbQTL) mapping analysis to identify host genetic variations associated with the abundance levels of bacterial taxa within the gut microbiome. Initially, 15 unknown bacteria were removed from 211 microbial datasets, resulting in the inclusion of 9 phyla, 16 classes, 20 orders, 32 families, and 119 genera [23]. Our study's selection of specific gut microbiota is based on MiBioGen consortium principles, highlighting the need to identify microbiota that significantly impact host metabolism, immunity, and disease. We concentrate on microbial species or communities proven to be prevalent in meta-analyses and linked to disease phenotypes through past GWAS findings.

The outcome database is sourced from proteins related to the pathogenesis of glioma(Dataset: prot-a-1217 in open gwas(https://gwas.mrcieu.ac.uk/datasets/prot-a-1217/), with glioma genetic data obtained from 3,301 individuals of European descent, comprising 10,534,735 SNPs (Fig 2).

## 2. Instrumental variable

Mendelian randomization is a method that uses genetic variations closely related to exposure factors as instrumental variables to establish whether there is a causal link between the exposure and a predetermined outcome. To ensure the accuracy and validity of the causal relationship between the gut microbiome and gliomas, single nucleotide polymorphisms (SNPs) must have a significant association with the exposure, fulfilling the assumption of relevance in Mendelian randomization. Our study employed two different statistical significance thresholds ($p<1\times10^{-5}$ and $p<5\times10^{-8}$) to select instrumental variables related to the gut microbiome. Due to the limited inclusion at $5\times10^{-8}$, we ultimately chose $1\times10^{-5}$ as the inclusion criterion. To ensure the reliability of the selected SNPs and eliminate bias, we performed linkage disequilibrium analysis ($R^2 <0.001$, clumping distance = 10,000kb) and detailed filtering of palindromic SNPs. To further validate the strong association between the instrumental variables and the

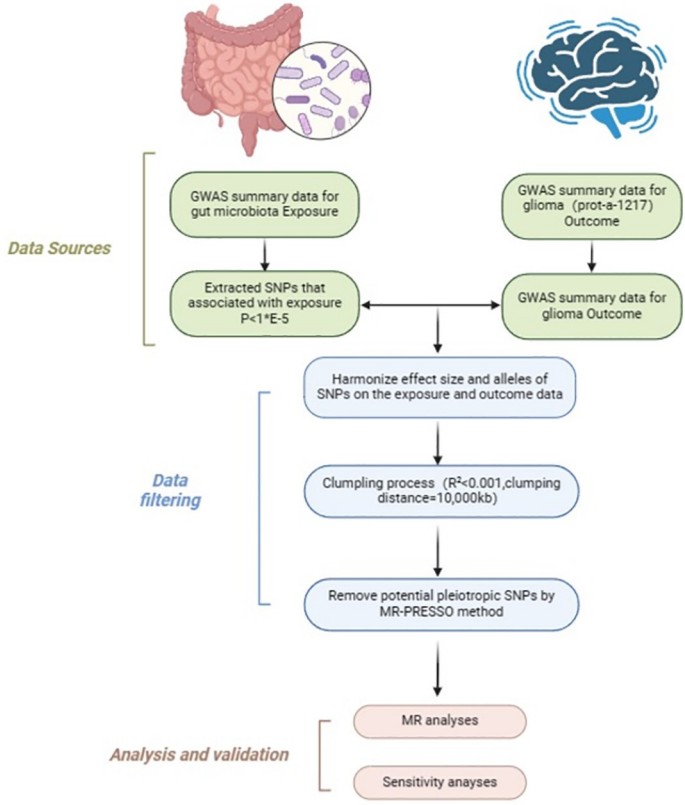

**Fig 2. This MR study is designed to investigate the association between gut microbiota and glioma.** Abbreviation: GWAS:whole genome association study; MR-PRESSO:Mendelian randomization pleiotropy;MR: Mendelian randomization; SNP: single nucleotide polymorphism.

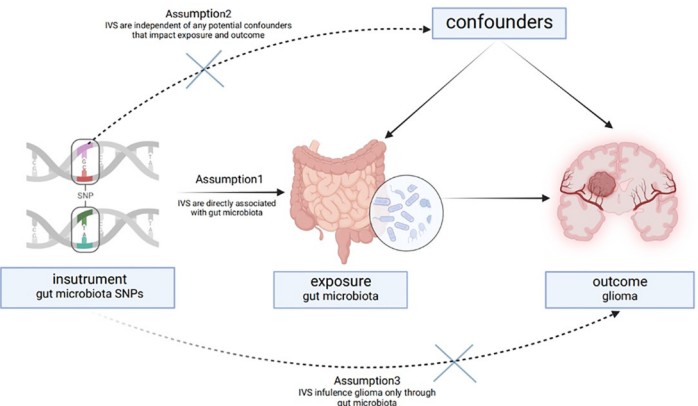

**Fig 3. Two-sample Mendelian randomized study design with genes predicting the effect of gut microbiota on gliomas.** IVS, instrumental variables; SNP, single nucleotide polymorphism.

exposure, we typically use F-statistics for quantitative assessment (Fig 3). An F-value below 10 indicates a weak association between the instrumental variable and the exposure.

## 3. Statistical analysis

In this study, we utilized protein data related to the pathogenesis of gliomas as our outcome data, sourced from the open GWAS database. To explore the causal relationship between the exposure factors and the outcome data, we employed Mendelian Randomization (MR) analysis. During the MR analysis, we adopted five widely-used methods, including MR Egger, Weighted Median, Inverse Variance Weighted (IVW), Simple Mode, and Weighted Mode. It's important to note that each MR analysis method has its own limitations. For instance, MR Egger is sensitive to outliers, while Weighted Median and Simple Mode may overlook other useful information. Ultimately, we chose the IVW method, one of the most commonly used in meta-analyses. This method integrates the effect size estimates from multiple studies through weighted averaging and yields unbiased results, assuming no horizontal pleiotropy. Lastly, to enhance the accuracy of our findings and reduce the false-positive rate, we applied the False Discovery Rate (FDR) method to adjust the P-values.

To comprehensively assess the stability and reliability of the MR analysis results obtained through the IVW method, we conducted a series of sensitivity analyses. These sensitivity tests primarily included in-depth analysis of genetic pleiotropy, detailed examination of heterogeneity, and the application of the leave-one-out method. We primarily used R Studio software (version 4.2) and the MR-presso package to carry out these sensitivity analyses.

## 4. Sensitivity analysis

In the analysis process, we initially conducted the Cochrane Q test, denoted as mr_heterogeneity. A Q_pvalue greater than 0.05 indicated the absence of significant heterogeneity in our sample, thereby boosting the credibility of our analysis results. Subsequently, in the MR-egger analysis, the intercept value P of the egger-intercept exceeded 0.05, further confirming the absence of pleiotropy issues. In other words, our MR analysis results are not influenced by confounding factors. Additionally, we employed the leave-one-out method, systematically excluding each Single Nucleotide Polymorphism (SNP) to check for any evident result bias. This step ensures that our findings are not dominated by one or a few specific SNPs, enhancing

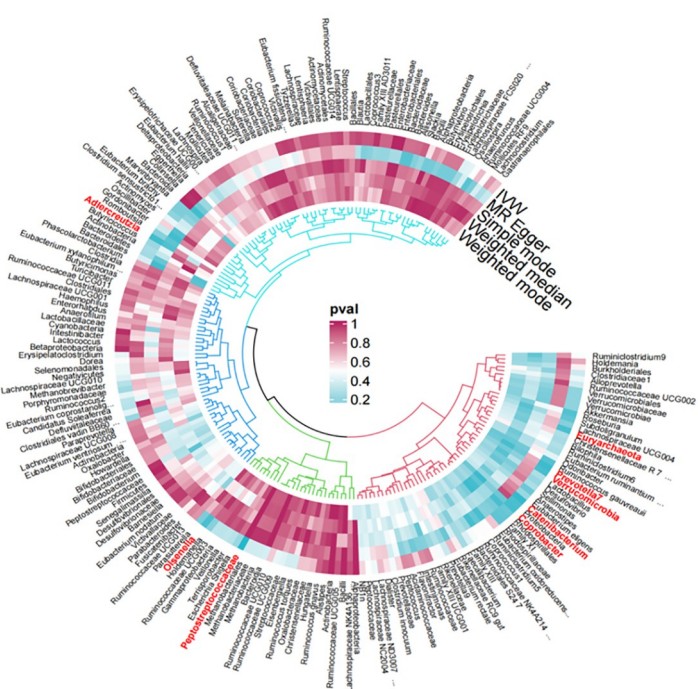

**Fig 4. 196 intestinal flora were visualized by MR analysis of 5 methods and outcome variables, and the red flora suggested a causal relationship.**

the robustness of the results (Fig 4). Unfortunately, when we attempted to further validate our conclusions using reverse Mendelian randomization studies, we found a lack of glioma-related datasets in the database. This limitation restricts our ability to comprehensively validate our findings, warranting more attention in future research.

## Result

After a series of quality control filters, we identified SNPs for inclusion based on a genome-wide significance threshold of $P < 1 \times 10^{-5}$. When the inclusion criterion was set at $P < 1 \times 10^{-8}$, fewer instrumental variables were available for analysis. To achieve a more comprehensive and unbiased result, we ultimately adopted $P < 1 \times 10^{-5}$ as the inclusion standard for this study. To further establish unbiased causal links, we obtained 92 SNPs as instrumental variables after linkage disequilibrium filtering and excluding SNPs related to outcome-associated confounding factors. The primary information of the included SNPs encompasses effect alleles, beta values, p-values, and standard errors (SE), which are used for further causal relationship analysis (S1 and S2 Files).

In our two-sample MR analysis, we used R language (version 4.2) to perform the analysis, employing IVW as the primary statistical method for evaluation. We identified eight microbial communities that may have a potential causal relationship with glioma (Fig 5). Specifically, four *genus Adlercreutzia* (OR = 1.25, 95%CI:1.02–1.54, P = 0.034, Q = 0.12), *Catenibacterium* (OR = 1.26, 95%CI:1.04–1.51, P = 0.017, Q = 0.98), *Coprobacter* (OR = 1.17, 95%CI:1.01–1.36, P = 0.041, Q = 0.29), and *Olsenella* (OR = 0.86, 95%CI:0.75–0.99, P = 0.048, Q = 0.44)—were identified. Among these, *Coprobacter* and *Olsenella* showed protective effects against glioma, while *Adlercreutzia* and *Catenibacterium* were identified as risk factors. From the 32 families, only one microbial community, *Peptostreptococcaceae* (OR = 0.67, 95%CI:0.57–0.86,

**Fig 5. Forest plot of causal relationship between eight flora and glioma.**

P = 0.001, Q = 0.34), showed a protective effect against glioma based on the 95% confidence interval of the OR value. *Verrucomicrobia* (OR = 1.27, 95%CI:1.01–1.57, P = 0.043, Q = 0.71), Prevotella7 (OR = 1.16, 95%CI:1.02–1.32, P = 0.029, Q = 0.61), and *Euryarchaeota* (OR = 1.13,95%CI:1.00–1.30, P = 0.049, Q = 0.66)wer identified as risk factors for glioma. (OR = 1.16,95%CI:1.02–1.32,P = 0.029,Q = 0.61),*Euryarchaeota*(OR = 1.13, 95%CI:1.00–1.30, P = 0.049, Q = 0.66)The Q-values, calculated using R language version 4.2 (via the mr_heterogeneity function in the TwoSampleMR package), indicate the results of heterogeneity tests. A Q-value greater than 0.05 suggests no heterogeneity and no bias in the results. To ensure the accuracy of our findings and minimize false positives, we applied FDR correction to the P-values. While our IVW estimates suggest some gut microbial communities have an implied association with glioma, these associations were no longer significant after FDR correction. After FDR correction, the absence of positive results might be related to sample size, experimental design, and potential confounding factors. Correlations with P < 0.05, but not reaching the FDR-controlled threshold, are considered to be suggestive (S3 File).

The F-value is used to validate the correlation between the instrumental variable and the exposure, calculated using the formula $F = R^2(n\text{-}k\text{-}1) / k(1\text{-}R^2)$. An F-value greater than 10 indicates a strong correlation between the instrumental variable and the exposure. Since we previously used IVW as the primary method for assessing the causal effect between gut microbiota and glioma, the IVW results can be considered unbiased only when it's proven that no pleiotropy exists between the instrumental variables and the outcome. Pleiotropy is detected using the mr_pleiotropy_test function in R's TwoSampleMR package (Table 1). Besides genetic pleiotropy, horizontal pleiotropy is also considered. It is tested using R language through the MR-presso package. When the MR-PRESSO results$Global Test$PvalueP > 0.05, it suggests that there is no horizontal pleiotropy between the exposure and outcome variables. Lastly, to ensure that no individual SNP has a significant impact on the outcome, we employed a leave-one-out approach, sequentially removing each SNP. Forest plots indicate that the overall error lines do not vary significantly, confirming the absence of obvious bias. Additionally, we plotted scatter graphs to analyze the risk and protective effects of each microbial community on glioma (S1 Fig).

**Table 1. Expression of OR and P values in five algorithms for eight flora.**

| Exposure | Outcome | Method | No of snp | P-value | OR | 95%CI |
|---|---|---|---|---|---|---|
| family.Peptostreptococcaceae | Glioma pathogenesis-related protein | MR Egger | 15 | 0.525 | 0.858 | 0.542–1.358 |
| | | Weighted median | 15 | 0.086 | 0.775 | 0.581–1.036 |
| | | IVW | 15 | 0.001 | 0.699 | 0.569–0.858 |
| | | Simple mode | 15 | 0.232 | 0.757 | 0.489–1.171 |
| | | Weighted mode | 15 | 0.248 | 0.794 | 0.546–1.155 |
| genus.Adlercreutzia | Glioma pathogenesis-related protein | MR Egger | 12 | 0.058 | 2.828 | 1.087–7.357 |
| | | Weighted median | 12 | 0.324 | 1.153 | 0.868–1.531 |
| | | IVW | 12 | 0.034 | 1.251 | 1.016–1.539 |
| | | Simple mode | 12 | 0.781 | 1.078 | 0.639–1.821 |
| | | Weighted mode | 12 | 0.824 | 1.055 | 0.662–1.681 |
| genus.Catenibacterium | Glioma pathogenesis-related protein | MR Egger | 5 | 0.833 | 1.217 | 0.225–6.568 |
| | | Weighted median | 5 | 0.144 | 1.207 | 0.946–1.541 |
| | | IVW | 5 | 0.018 | 1.253 | 1.039–1.511 |
| | | Simple mode | 5 | 0.346 | 1.177 | 0.869–1.593 |
| | | Weighted mode | 5 | 0.385 | 1.174 | 0.847–1.627 |
| genus.Coprobacter | Glioma pathogenesis-related protein | MR Egger | 14 | 0.111 | 1.468 | 0.948–2.273 |
| | | Weighted median | 14 | 0.025 | 1.269 | 1.029–1.564 |
| | | IVW | 14 | 0.041 | 1.168 | 1.005–1.357 |
| | | Simple mode | 14 | 0.158 | 1.311 | 0.919–1.871 |
| | | Weighted mode | 14 | 0.115 | 1.285 | 0.960–1.722 |
| genus.Olsenella | Glioma pathogenesis-related protein | MR Egger | 11 | 0.801 | 1.077 | 0.612–1.897 |
| | | Weighted median | 11 | 0.553 | 0.941 | 0.769–1.154 |
| | | IVW | 11 | 0.048 | 0.862 | 0.745–0.998 |
| | | Simple mode | 11 | 0.841 | 0.969 | 0.709–1.327 |
| | | Weighted mode | 11 | 0.831 | 0.967 | 0.723–1.293 |
| phylum.Verrucomicrobia | Glioma pathogenesis-related protein | MR Egger | 12 | 0.722 | 1.226 | 0.603–2.089 |
| | | Weighted median | 12 | 0.129 | 1.263 | 0.934–1.707 |
| | | IVW | 12 | 0.043 | 1.258 | 1.001–1.574 |
| | | Simple mode | 12 | 0.181 | 1.439 | 0.871–2.381 |
| | | Weighted mode | 12 | 0.179 | 1.399 | 0.891–2.197 |
| genus Prevotella7 | Glioma pathogenesis-related protein | MR Egger | 11 | 0.692 | 1.178 | 0.537–2.179 |
| | | Weighted median | 11 | 0.038 | 1.211 | 1.005–1.461 |
| | | IVW | 11 | 0.029 | 1.159 | 1.016–1.321 |
| | | Simple mode | 11 | 0.123 | 1.329 | 0.969–1.821 |
| | | Weighted mode | 11 | 0.121 | 1.328 | 0.959–1.842 |
| phylum Euryarchaeota | Glioma pathogenesis-related protein | MR Egger | 12 | 0.664 | 0.876 | 0.491–1.562 |
| | | Weighted median | 12 | 0.141 | 1.137 | 0.958–1.349 |
| | | IVW | 12 | 0.049 | 1.132 | 1.001–1.304 |
| | | Simple mode | 12 | 0.397 | 1.147 | 0.867–1.521 |
| | | Weighted mode | 12 | 0.401 | 1.138 | 0.871–1.487 |

## Discussion

The gut microbiota plays a crucial role in human physiology, particularly in regulating and maintaining the immune system [20, 24].These microbial communities are influenced by various factors, including the host's dietary habits, gastrointestinal health, and antibiotic use, all of which can disrupt the microbial balance. The activation or suppression of the immune system and inflammatory responses not only affects the progression of multiple diseases but also alters

the risk of specific conditions like cancer. In cancer research, an imbalance in gut microbiota can trigger inflammatory responses, subsequently affecting cellular growth and division pathways, making it a key subject for assessing cancer risk [25].

In neuro-oncology, particularly in glioma research, tumor growth and the immune-suppressive characteristics unique to gliomas impact the body's natural anti-tumor responses. For instance, glioma cells elevate levels of immune-suppressive factors like PD-L1 and Indoleamine 2,3-dioxygenase (IDO), thereby limiting antigen presentation [26, 27]. Additionally, macrophages associated with gliomas secrete IL-10 and Transforming Growth Factor-β (TGF-β), further dampening immune cell activity [28, 29]. These elements collectively create an immune-suppressive tumor microenvironment, neutralizing the host's natural defense mechanisms [30]. Regulatory T (Treg) cells in the glioma microenvironment further enhance immune suppression by depleting cytotoxic T lymphocytes, directly compromising tumor cells [31]. Animal studies have shown that antibiotic-induced changes in gut microbiota can impair NK cell function, weaken their direct cytotoxic effects on cancer cells, and induce changes in microglial cell phenotypes [32]. These findings further confirm the pivotal role of the Gut-Brain axis in regulating bodily responses, suggesting that gut microbiota may promote the formation of a tumor-tolerant microenvironment, thereby facilitating further tumor development.

Metabolomic analysis of the gut has revealed certain metabolites that may play a significant role in immune responses [33], Our study, which analyzed 32 different bacterial families, found a potential protective effect between *Peptostreptococcaceae* and gliomas. This finding contradicts previous RCT studies on gut metabolomics, where higher cancer incidence was observed in populations with a higher prevalence of this bacterial family in their feces. However, no specific studies have yet explored the relationship between this bacterial family and gliomas, making further investigation into their causal relationship clinically significant. *Adlercreutzia*, a genus of bacteria, is considered a potential risk factor for gliomas in our study. Although clinical studies on this bacterial group are limited, animal experiments have shown its effectiveness in regulating inflammatory responses in the body [34]. While there's no definitive evidence linking this bacterial group directly to gliomas, it's reasonable to hypothesize that it may influence the onset and progression of gliomas through its role in regulating inflammation.

The *Prevotella* genus in our study activates Toll-like receptor 2 to promote antigen-presenting cells to produce Th17-polarizing cytokines, including Interleukin-23 (IL-23) and IL-1. This bacterial group induces mucosal inflammation, leading to the production of inflammatory mediators that could elevate the risk of systemic diseases and cancer, aligning with our study's conclusions [35]. Immune checkpoint blockade (ICB) therapies, which activate the therapeutic potential of the immune system, have proven effective for certain types of tumors and cancer patients [36]. In our study, the gut bacteria *Olsenella* and *Coprobacter* were negatively correlated with the risk of gliomas. Previous animal studies on these bacterial groups have shown that they significantly enhance the efficacy of ICB therapies, a finding consistent with our study. This is achieved by modulating CTLA-4 or PD-1 and their ligand PD-L1 [37, 38]. A discovery that has had a profound impact on the treatment of certain types of cancer. While these preliminary findings align with the main conclusions of our paper, further clinical studies are needed to confirm whether these bacterial groups can be used in the treatment of gliomas. Literature on the *Catenibacterium* genus suggests a correlation with neurological diseases, giving us reason to speculate that this bacterial group may influence the development of brain diseases through certain mechanisms. However, due to limited research on *Verrucomicrobia* and *Euryarchaeota*, this paper will not discuss these bacterial groups.

One significant strength of this study is its use of the largest available sample size to explore the relationship between gut microbiota and gliomas, effectively eliminating confounding factors and individual variations from a statistical standpoint. However, the study has several

limitations. First, due to data constraints, we were unable to perform reverse Mendelian Randomization (MR) tests, preventing us from confirming the bidirectional effects of the Gut-Brain axis. Second, although MR is a hypothesis-driven method suitable for detecting causal relationships when some biological evidence exists, most of our results have not undergone rigorous False Discovery Rate (FDR) correction. Third, the selection of instrumental variables could be influenced by weak instruments. While we verified the strong correlation of our chosen instruments through F-values, their low sensitivity warrants cautious interpretation. Lastly, the glioma data used in this study were not categorized in detail, preventing us from conducting subgroup analyses and, consequently, from delving deeper into the causal relationships between different types of gliomas and gut microbiota.

## Conclusion

In summary, by utilizing publicly available genetic databases, we've established a clear causal relationship between gut microbiota and gliomas. However, to fully understand the complex interplay between the two, further statistical analyses and clinical validations are needed. Based on existing scientific evidence, we can reasonably infer the potential pathways through which these relevant microbial communities may regulate the development and progression of gliomas. These findings not only offer valuable directions for future clinical research but also present new perspectives for the formulation and optimization of drug treatment strategies. Overall, these preliminary results provide new avenues for research and potential treatments to address the serious medical issue of gliomas.

## Supporting information

**S1 Fig.**
(PNG)

**S1 File. Details about the selected instrumental variables are shown in Additional file 1: S1 Table.**
(TXT)

**S2 File. $R^2$ and F values for the positive microbiota instrumental variables.**
(CSV)

**S3 File. Assessment of directional level pleiotropy in MR Egger regression by intercept term associated between gut microbiota and glioma.**
(CSV)

**S1 Appendix.**
(PDF)

**S2 Appendix.**
(PDF)

**S3 Appendix.**
(PDF)

**S4 Appendix.**
(PDF)

**S5 Appendix.**
(PDF)

**S6 Appendix.**
(PDF)

**S7 Appendix.**
(PDF)

**S8 Appendix.**
(PDF)

**S9 Appendix.**
(PDF)

**S10 Appendix.**
(PDF)

**S11 Appendix.**
(PDF)

**S12 Appendix.**
(PDF)

**S13 Appendix.**
(PDF)

**S14 Appendix.**
(PDF)

**S15 Appendix.**
(PDF)

**S16 Appendix.**
(PDF)

## Acknowledgments

We gratefully acknowledge the authors and participants of all GWASs and MiBioGen from which we used summary statistics data.

## Author Contributions

**Conceptualization:** Chenzhi Cui, Tianke Yang, Zhuqiang Jia, Xin Han, Shouyu Wang.

**Data curation:** Tianke Yang, Zhuqiang Jia, Xin Han, Shouyu Wang.

**Formal analysis:** Tianke Yang, Zhuqiang Jia, Xin Han.

**Funding acquisition:** Shouyu Wang.

**Investigation:** Tianke Yang, ShengYu Wang, Zhuqiang Jia, Xin Han, Dong Chen.

**Methodology:** Tianke Yang, ShengYu Wang, Zhuqiang Jia, Dong Chen.

**Project administration:** Tianke Yang, ShengYu Wang, Dong Chen.

**Resources:** Tianke Yang, ShengYu Wang, Lin Zhao, Xiaohong Sun, Junwei Zong, Shouyu Wang, Dong Chen.

**Software:** Tianke Yang, ShengYu Wang, Lin Zhao, Xiaohong Sun, Junwei Zong.

**Supervision:** Tianke Yang, ShengYu Wang, Lin Zhao, Xiaohong Sun, Junwei Zong.

**Validation:** Tianke Yang, Lin Zhao, Junwei Zong.

**Visualization:** Tianke Yang, Junwei Zong, Shouyu Wang.

**Writing – original draft:** Tianke Yang.

**Writing – review & editing:** Chenzhi Cui.

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
