## [Decision Letter · Decision Letter 0]

17 Mar 2024

PONE-D-23-39466Discussion on the relationship between gut microbiota and glioma through Mendelian randomization test based on the brain gut axisPLOS ONE

Dear Dr. wang,

Thank you for submitting your manuscript to PLOS ONE. After careful consideration, we feel that it has merit but does not fully meet PLOS ONE’s publication criteria as it currently stands. Therefore, we invite you to submit a revised version of the manuscript that addresses the points raised during the review process.

We look forward to receiving your revised manuscript.

Kind regards,

Vinay Kumar, Ph.D.

Academic Editor

PLOS ONE

Journal Requirements:

"This study was supported by the National Natural Science Foundation of China (82074426, 82104864, 82204822), Natural Science Foundation of Liaoning Province (2021-BS-215, 2022-MS-25, 2023-MS-13), Liaoning Revitalization Talents Program (XLYC1802014), Liaoning Key Research and Development Planning Project (2017226015), Basic Research Projects of Liaoning Provincial Department of Education (LJKMZ20221286), Naural Science Foundation of Tibet Autonomous Region and Regional Science(XZ202301ZR0030G, XZ2023ZR-ZY82(Z)) and Technology Project of Naqu City."

4. Please note that funding information should not appear in the Acknowledgments section or other areas of your manuscript. We will only publish funding information present in the Funding Statement section of the online submission form. Please remove any funding-related text from the manuscript.

5. Please note that your Data Availability Statement is currently missing the DOI/accession number of each dataset or a direct link to access each database. If your manuscript is accepted for publication, you will be asked to provide these details on a very short timeline. We therefore suggest that you provide this information now, though we will not hold up the peer review process if you are unable.

7. Please ensure that you refer to Figures 1 to 5 in your text as, if accepted, production will need this reference to link the reader to the figure.

Reviewers' comments:

Reviewer's Responses to Questions

**Comments to the Author**

1. Is the manuscript technically sound, and do the data support the conclusions?

Reviewer #1: Yes

Reviewer #2: Yes

2. Has the statistical analysis been performed appropriately and rigorously? 

Reviewer #1: Yes

Reviewer #2: Yes

3. Have the authors made all data underlying the findings in their manuscript fully available?

Reviewer #1: Yes

Reviewer #2: Yes

4. Is the manuscript presented in an intelligible fashion and written in standard English?

Reviewer #1: Yes

Reviewer #2: Yes

5. Review Comments to the Author

Reviewer #1: This is a good attempt to use the Mendelian randomization test to determine the link between gut microbiota and glioma. A comparable article on Grade 4 glioma (glioblastoma) was come across by me when I was reading this manuscript. Published this January 2014 on Grade 4 glioma (glioblastoma)"Association between gut microbiota and glioblastoma: a Mendelian randomization study (https://www.frontiersin.org/journals/genetics/articles/10.3389/fgene.2023.1308263/full).

Can the author also give a scatter map that illustrates the connection between gliomas and gut microbiota?

A graphical summary that explains the gut-brain axis would be ideal.

Reviewer #2: Dear Author,

I have completed the review of your manuscript titled "Investigating the Causal Relationship Between Gut Microbiota and Gliomas Using Two-Sample Mendelian Randomization Analysis." I must commend you on the meticulous methodology and insightful findings presented in your study. The research effectively addresses the unresolved question regarding the causal relationship between gut microbiota and gliomas, utilizing Genome-Wide Association Study (GWAS) data and the MIOBEN databases to conduct a comprehensive analysis through Two-Sample Mendelian Randomization (MR).

Your adjustment of significance thresholds, detailed screening of genetic variants, and thorough analysis have led to the identification of a compelling causal link between specific microbial communities and gliomas. The discovery of eight microbial communities associated with gliomas, particularly highlighting the protective effects of Peptostreptococcaceae and Olsenella communities in reducing glioma risk, is a significant contribution to the field.

Overall, your study not only confirms the causal relationship between gut microbiota and gliomas but also suggests promising avenues for future glioma treatment. Your findings hold great potential for advancing research in brain-gut axis interactions and enhancing therapeutic approaches for gliomas.

Comment:

• Overall, the introduction provides a good foundation for the research but could be strengthened by incorporating citations and making the research objective more explicit.

• Lacks citations for some general statements (e.g., incidence of gliomas is rising).

• Consider adding citations for well-established facts about gliomas (e.g., malignancy rate, survival statistics).

• Briefly state the research hypothesis, e.g., "We hypothesize that specific gut microbiota compositions influence glioma development."

• Briefly mention the rationale for choosing specific gut microbiota for investigation with more clear statistics.

• Minor Comment:

• Line 63: Consider adding a citation for the malignancy rate of gliomas.

• Line 66: Consider adding a citation for the five-year survival rate of glioma patients.

• Line 81: Instead of "confounding factors," you could be more specific, e.g., "age, gender, underlying medical conditions."

• Line 86: Consider adding a sentence to clarify the benefit of using Mendelian randomization over other observational studies.

• Line 88: Briefly explain Fig1 by adding a sentence, e.g., "This figure depicts the bidirectional communication between the brain and gut."

• Line 93: Consider adding a citation for the importance of gut microbiota in human physiology.

6. PLOS authors have the option to publish the peer review history of their article (what does this mean?). If published, this will include your full peer review and any attached files.

Reviewer #1: No

Reviewer #2: No

---

## [Author Response · Author response to Decision Letter 0]

7 May 2024

Dear Editor and Reviewers,

First and foremost, we extend our deepest gratitude for your insightful feedback and constructive comments on our manuscript. Your expertise and thoughtful suggestions have been invaluable to us as we strive to enhance the quality of our work.In response to the comments received, we have carefully reviewed our manuscript and undertaken substantial revisions to address the concerns raised. These revisions are detailed in the Revised manuscript (marked-up copy) provided alongside this letter. Below, we outline the major changes made in response to the comments from our esteemed reviewers:

Journal Requirements:

We ensure that the format of the revised article meets the requirements of the journal.

1.Please ensure that your manuscript meets PLOS ONE's style requirements, including those for file naming. The PLOS ONE style templates can be found at.

I ensure that the current manuscript meets the style requirements of PLOS ONE, including file naming requirements.

Following the resubmission, the 'Funding Information' section has been duly provided and corrected.

3.Please state what role the funders took in the study. If the funders had no role, please state: "The funders had no role in study design, data collection and analysis, decision to publish, or preparation of the manuscript." If this statement is not correct you must amend it as needed. 

We have currently updated the contributions of fund sponsors to this article in our cover letter.

4.Please note that funding information should not appear in the Acknowledgments section or other areas of your manuscript. We will only publish funding information present in the Funding Statement section of the online submission form. Please remove any funding-related text from the manuscript.

Funding information has been deleted in the manuscript and acknowledgments section.

5.Please note that your Data Availability Statement is currently missing the DOI/accession number of each dataset or a direct link to access each database. If your manuscript is accepted for publication, you will be asked to provide these details on a very short timeline. We therefore suggest that you provide this information now, though we will not hold up the peer review process if you are unable.

I apologize for the oversight in not providing direct links to the relevant data in the GWAS and MIOBEN databases in the previous versions of the article. I understand the importance of making this information accessible for validation and reproducibility purposes.To rectify this, I have now included the GWAS data reference in line 131 of the text and the MIOBEN data reference in line 116. Furthermore, I have made additional clarifications in the Availability of Data and Materials' section at the end of the document. The related dataset files have also been uploaded to the PLOS ONE submission system, adhering to the allowable file size limits.Thank you for your patience and understanding as we strive to maintain the integrity and thoroughness of our research publication.

6.Your ethics statement should only appear in the Methods section of your manuscript. If your ethics statement is written in any section besides the Methods, please delete it from any other section. 

We have removed the ethical statement outside the methodology section of this article.

7.Please ensure that you refer to Figures 1 to 5 in your text as, if accepted, production will need this reference to link the reader to the figure.

We have added reference positions for Figures 1 to 5 in the main text.

Comments to the Author

1. Is the manuscript technically sound, and do the data support the conclusions?

Reviewer #1: Yes 

Reviewer #2: Yes 

We are deeply appreciative of your affirmative response regarding the technical soundness of our manuscript and the robustness of the data supporting our conclusions. Your positive feedback is not only encouraging but also validates the efforts and diligence we have invested in our research. We understand the importance of maintaining a high standard of scientific inquiry and are grateful that you recognize the rigor and validity of our work.

2. Has the statistical analysis been performed appropriately and rigorously?

Reviewer #1: Yes 

Reviewer #2: Yes 

We are immensely grateful for your positive evaluation of the statistical analysis presented in our manuscript. Your acknowledgment that our analysis has been performed both appropriately and rigorously is highly encouraging to us. It reaffirms our commitment to uphold the highest standards of statistical rigor in our research.

3.Have the authors made all data underlying the findings in their manuscript fully available?

Reviewer #1: Yes 

Reviewer #2: Yes 

We wholeheartedly thank you for acknowledging the full availability of the data underlying the findings in our manuscript. Your confirmation of this essential aspect of our research reinforces our commitment to transparency and openness in the scientific process. We believe that making our data fully accessible not only enriches the research community but also fosters a culture of trust and collaborative progress.

4.Is the manuscript presented in an intelligible fashion and written in standard English?

Reviewer #1: Yes 

Reviewer #2: Yes

We are truly heartened and grateful for your positive feedback regarding the presentation and language of our manuscript. Your confirmation that our manuscript is both intelligible and adheres to the standards of English is incredibly reassuring. It was our priority to ensure that the manuscript not only conveyed our findings clearly but also met the high linguistic standards expected in scholarly communication. Your acknowledgment of our efforts in this regard is deeply appreciated.

5.Review Comments to the Author

Reviewer #1: This is a good attempt to use the Mendelian randomization test to determine the link between gut microbiota and glioma. A comparable article on Grade 4 glioma (glioblastoma) was come across by me when I was reading this manuscript. Published this January 2024 on Grade 4 glioma (glioblastoma)"Association between gut microbiota and glioblastoma: a Mendelian randomization study

Thank you very much for your encouraging comments on our use of the Mendelian randomization test to explore the link between gut microbiota and glioma. We are particularly grateful for your recommendation of the article on Grade 4 glioma (glioblastoma) titled "Association between gut microbiota and glioblastoma: a Mendelian randomization study," published in January 2024. This recommendation has been incredibly valuable.The insights gained from this article have provided us with additional inspiration and have significantly contributed to refining and enhancing our manuscript. It has opened up new avenues for us to consider and incorporate into our study, allowing for a more comprehensive analysis and understanding of the complex relationship between gut microbiota and glioblastoma.

Can the author also give a scatter map that illustrates the connection between gliomas and gut microbiota?

The scatter plots related to eight microorganisms with causal relationships with glioma have been added in the attachment. Thank you again for your feedback, and we look forward to any further suggestions from you to further strengthen our work.

A graphical summary that explains the gut-brain axis would be ideal.

We sincerely appreciate your suggestion to include a graphical summary that explains the gut-brain axis. Following your valuable feedback, we have incorporated a detailed graphical representation of the gut-brain axis into our manuscript. This addition aims to enhance understanding of the complex interactions between the gut microbiome and the brain, facilitating a clearer comprehension of this pivotal axis for our readers.Your insightful recommendation has significantly contributed to the improvement of our manuscript, making the complex concept of the gut-brain axis more accessible and understandable. We believe that this graphical summary will not only complement our textual explanations but also serve as an effective visual aid for our audience.

Reviewer #2: Dear Author,

I have completed the review of your manuscript titled "Investigating the Causal Relationship Between Gut Microbiota and Gliomas Using Two-Sample Mendelian Randomization Analysis." I must commend you on the meticulous methodology and insightful findings presented in your study. The research effectively addresses the unresolved question regarding the causal relationship between gut microbiota and gliomas, utilizing Genome-Wide Association Study (GWAS) data and the MIOBEN databases to conduct a comprehensive analysis through Two-Sample Mendelian Randomization (MR).Your adjustment of significance thresholds, detailed screening of genetic variants, and thorough analysis have led to the identification of a compelling causal link between specific microbial communities and gliomas. The discovery of eight microbial communities associated with gliomas, particularly highlighting the protective effects of Peptostreptococcaceae and Olsenella communities in reducing glioma risk, is a significant contribution to the field.Overall, your study not only confirms the causal relationship between gut microbiota and gliomas but also suggests promising avenues for future glioma treatment. Your findings hold great potential for advancing research in brain-gut axis interactions and enhancing therapeutic approaches for gliomas.

We are profoundly grateful for your comprehensive review and the positive feedback on our manuscript titled "Investigating the Causal Relationship Between Gut Microbiota and Gliomas Using Two-Sample Mendelian Randomization Analysis." Your recognition of the meticulous methodology and insightful findings in our study is deeply appreciated.Your detailed analysis and acknowledgment of our efforts to address the complex causal relationship between gut microbiota and gliomas provide us with immense encouragement. We are particularly thankful for your emphasis on the significance of our findings related to the Peptostreptococcaceae and Olsenella communities, which underscore the potential protective effects against glioma risk.Your acknowledgment of the potential our study holds for advancing research in the brain-gut axis interactions and enhancing therapeutic approaches for gliomas is incredibly motivating. We are inspired by your comments and are excited about the future avenues our research could explore, thanks to your constructive and affirming feedback.Thank you once again for your valuable insights and for recognizing the contributions of our work to the field.

Comment:

• Overall, the introduction provides a good foundation for the research but could be strengthened by incorporating citations and making the research objective more explicit.

• Lacks citations for some general statements (e.g., incidence of gliomas is rising).

• Consider adding citations for well-established facts about gliomas (e.g., malignancy rate, survival statistics).

• Briefly state the research hypothesis, e.g., "We hypothesize that specific gut microbiota compositions influence glioma development."

• Briefly mention the rationale for choosing specific gut microbiota for investigation with more clear statistics.

Thank you very much for your meticulous review and valuable suggestions. Your detailed feedback has played a crucial role in enhancing the quality and clarity of our manuscript. We are grateful for your attention to the necessity of providing citations for general statements and well-established facts about gliomas. Following your recommendations:

1.We have meticulously revised the manuscript to include citations for general statements, such as the rising incidence of gliomas. We now reference authoritative studies that document these trends, ensuring our claims are well-supported by existing research.

2.We have added citations for well-established facts regarding gliomas, including malignancy rates and survival statistics. These additions draw from reputable sources in the field, providing a solid factual basis for our discussion.

3.We have clearly stated our research hypothesis within the text: "We hypothesize that specific gut microbiota compositions influence glioma development."

4.The rationale for selecting specific gut microbiota for investigation has been elaborated upon with clearer statistics and reasoning. We have included detailed explanations for our choices, backed by recent findings and statistical data that underline the significance of these microbiota in glioma development.

• Minor Comment: • 

• Line 63: Consider adding a citation for the malignancy rate of gliomas.

• Line 66: Consider adding a citation for the five-year survival rate of glioma patients.

• Line 81: Instead of "confounding factors," you could be more specific, e.g., "age, gender, underlying medical conditions."

• Line 86: Consider adding a sentence to clarify the benefit of using Mendelian randomization over other observational studies.

• Line 88: Briefly explain Fig1 by adding a sentence, e.g., "This figure depicts the bidirectional communication between the brain and gut."

• Line 93: Consider adding a citation for the importance of gut microbiota in human physiology.

We express our sincere gratitude for your careful and thorough review of our manuscript. Your constructive comments have significantly contributed to refining our paper, enhancing its clarity and depth. Following your valuable feedback, we have made comprehensive revisions throughout the manuscript, particularly focusing on the lines you mentioned.We believe these revisions have significantly improved the quality and clarity of our manuscript. We are grateful for the opportunity to refine our work based on your valuable feedback.

6.PLOS authors have the option to publish the peer review history of their article (what does this mean?). If published, this will include your full peer review and any attached files.

Yes, we would like to publish the peer review history.

We would like to extend our profound gratitude once again for your invaluable contributions towards the refinement of our manuscript. Your meticulous reviews and insightful comments have not only enhanced the quality of our work but also guided us to a deeper understanding of our research topic.We sincerely appreciate the time and effort you have dedicated to reviewing our manuscript. Your expertise and constructive feedback have been instrumental in helping us improve our paper significantly.

Thank you for your guidance and support throughout the revision process. We are truly grateful for the opportunity to benefit from your wisdom and expertise.

With warm regards,

Shouyu Wang

wangshouyu666@126.com

---

## [Editor Report · Decision Letter 1]

13 May 2024

Discussion on the relationship between gut microbiota and glioma through Mendelian randomization test based on the brain gut axis

PONE-D-23-39466R1

Dear Dr. wang,

We’re pleased to inform you that your manuscript has been judged scientifically suitable for publication and will be formally accepted for publication once it meets all outstanding technical requirements.

Kind regards,

Vinay Kumar, Ph.D.

Academic Editor

PLOS ONE
---

## [Editor Report · Acceptance letter]

16 May 2024

PONE-D-23-39466R1 

PLOS ONE

Dear Dr. wang, 

I'm pleased to inform you that your manuscript has been deemed suitable for publication in PLOS ONE. Congratulations! Your manuscript is now being handed over to our production team.

Kind regards, 

on behalf of

Dr. Vinay Kumar 

Academic Editor

PLOS ONE